# Multiplex RT Real-Time PCR Based on Target Failure to Detect and Identify Different Variants of SARS-CoV-2: A Feasible Method That Can Be Applied in Clinical Laboratories

**DOI:** 10.3390/diagnostics13081364

**Published:** 2023-04-07

**Authors:** Van Hung Pham, Huong Thien Pham, Mario G. Balzanelli, Pietro Distratis, Rita Lazzaro, Quoc Viet Nguyen, Viet Quoc Tran, Duy Khanh Tran, Luan Duy Phan, Sang Minh Pham, Binh Thai Pham, Chien Vo Duc, Ha Minh Nguyen, Dung Ngoc Thi Nguyen, Ngoc Van Tran, Son Truong Pham, Camelia Queck, Kieu Diem Cao Nguyen, Francesco Inchingolo, Raffaele Del Prete, Nam Hai Dinh Nguyen, Luigi Santacroce, Ciro Gargiulo Isacco

**Affiliations:** 1Department of Microbiology, Phan Chau Trinh University, Dien Ban 550000, Vietnam; 2International Research Institute of Gene and Immunology, Ho Chi Minh City 700000, Vietnam; 3SET-118, Department of Pre-Hospital and Emergency, SG Giuseppe Moscati Hospital, 74010 Taranto, Italy; 4Nam Khoa Co., Ltd., Ho Chi Minh City 700000, Vietnam; 5Nguyen Tri Phuong Hospital, Ho Chi Minh City 700000, Vietnam; 6HCMC Society of Medicine, Ho Chi Minh City 700000, Vietnam; 7New South Wales Health, Sydney 2065, Australia; 8Faculty of Medicine and Health, The University of Sydney, Sydney 2006, Australia; 9Department of Interdisciplinary Medicine, Section of Dentistry, Microbiology and Virology, School of Medicine, University of Bari “Aldo Moro”, 70121 Bari, Italy

**Keywords:** SARS-CoV-2, Delta variant, Omicron variant, multiplex reverse transcriptase real-time PCR, target-failure real-time PCR

## Abstract

Shortly after its emergence, Omicron and its sub-variants have quickly replaced the Delta variant during the current COVID-19 outbreaks in Vietnam and around the world. To enable the rapid and timely detection of existing and future variants for epidemiological surveillance and diagnostic applications, a robust, economical real-time PCR method that can specifically and sensitively detect and identify multiple different circulating variants is needed. The principle of target- failure (TF) real-time PCR is simple. If a target contains a deletion mutation, then there is a mismatch with the primer or probe, and the real-time PCR will fail to amplify the target. In this study, we designed and evaluated a novel multiplex RT real-time PCR (MPL RT-rPCR) based on the principle of target failure to detect and identify different variants of SARS-CoV-2 directly from the nasopharyngeal swabs collected from COVID-19 suspected cases. The primers and probes were designed based on the specific deletion mutations of current circulating variants. To evaluate the results from the MPL RT-rPCR, this study also designed nine pairs of primers for amplifying and sequencing of nine fragments from the S gene containing mutations of known variants. We demonstrated that (i) our MPL RT-rPCR was able to accurately detect multiple variants that existed in a single sample; (ii) the limit of detection of the MPL RT-rPCR in the detection of the variants ranged from 1 to 10 copies for Omicron BA.2 and BA.5, and from 10 to 100 copies for Delta, Omicron BA.1, recombination of BA.1 and BA.2, and BA.4; (iii) between January and September 2022, Omicron BA.1 emerged and co-existed with the Delta variant during the early period, both of which were rapidly replaced by Omicron BA.2, and this was followed by Omicron BA.5 as the dominant variant toward the later period. Our results showed that SARS-CoV-2 variants rapidly evolved within a short period of time, proving the importance of a robust, economical, and easy-to-access method not just for epidemiological surveillance but also for diagnoses around the world where SARS-CoV-2 variants remain the WHO’s highest health concern. Our highly sensitive and specific MPL RT-rPCR is considered suitable for further implementation in many laboratories, especially in developing countries.

## 1. Introduction

Since the onset of COVID-19, SARS-CoV-2 virus has undergone a number of variants, including four variants of concern: Alpha, Beta, Gamma, and Delta. From November 2021 onward, these variants have gradually disappeared and been replaced by the Omicron variant, which was initially reported in South Africa and later worldwide with an extremely high transmission rate [1,2]. However, the Omicron variant is not stable and it has been continuing to evolve. To date, according to the covariant.org with data enabled by the GISAID, eight sub-variants of Omicron have been reported [3], including BA.1 (21K), BA.2 (21L), BA.4 (22A), BA.5 (22B), BA.2.12.1 (22C), BA.2.75 (22D), BQ.1 (22E), and XBB (22F). To detect and identify different variants of SARS-CoV-2 from collected samples, investigators often use genome sequencing with next-generation sequencing technology [4]. This technique has the advantage of not only detecting and distinguishing variants of SARS-CoV-2 virus but also being able to monitor variation in the SARS-CoV-2 genome, thereby making it possible to detect new variants [4]. However, next-generation sequencing is awfully expensive and difficult to use in local monitoring laboratories, especially in countries where funding is not plentiful enough to achieve the advantages of this technology. In Vietnam, during the last pandemic wave (July to October 2021), Delta variants were reported as the main causative pathogens. However, in December, the Omicron variants started to be reported. Because of the high cost of genome sequencing, the detection and identification of SARS-CoV-2 in Vietnam is limited to samples with high viral load [4] and cannot reflect the real distribution of existing variants. This issue has prompted us to develop the multiplex reverse transcriptase real-time PCR (MPL RT-rPCR) to detect the different variants existing in samples collected from COVID-19 suspected cases, which in turn can help find out the distribution of these variants in the current increasingly contagious epidemic situation.

## 2. Aims of the Study

The main aim of the study is to use and evaluate the multiplex RT-rPCR (MPL RT-rPCR) based on the principle of target failure to detect and identify current circulating SARS-CoV-2 variants directly from the nasopharyngeal swabs taken from COVID-19 suspected individuals. The specific objectives include the following: (1) to determine the accuracy of the MPL RT-rPCR through a comparison with Sanger sequencing of the mutation-containing regions on the S gene of the variants; (2) to determine the limit of detection (LOD) of the MPL RT-rPCR in detecting the different variants existing in the tested samples; and (3) to understand the time of occurrence as well as the distribution rate of current circulating variants in Ho Chi Minh City during the period of study via analyzing the results coming from the MPL RT-rPCR.

## 3. Materials and Study Methods

### 3.1. Primers and Probes for the Detection of SARS-CoV-2 and Its Variants

For the detection of SARS-CoV-2, the primers and probes recommended by CDC to detect the N2 fragment of the N gene were selected since they targeted the conserved regions on the N gene without any reported mutations to date. The primers and probes targeting the SARS-CoV-2 variants were designed to detect the deletion mutations that might be specific for the variants. The principle of action of these primers and probes for the detection of deletion mutations is based on the target-failure real-time PCR: if a target contains a deletion mutation, then there is a mismatch with the primer or probe and the real-time PCR will fail to amplify the target. Besides those specific for SARS-CoV-2 and its variants, the primers and probes targeting the human RNAse P were also used as the internal controls (IC). All the primes and probes were set up in two MPL RT-rPCR, with **MPL1** targeting **S:** 142-144 del, **S:** 69-70 del, **S:** N211 del, and **S:** 24-26 del, and **MPL2** targeting **S:** 156-157 del, **ORF1a:** 141-143 del, **N2** of SARS-CoV-2, and **RNAseP**. Table 1 shows the sequences of the primers and probes in the MPL1 and MPL2 and its targets.

The MPL RT-rPCR mixes of MPL1 and MPL2 were prepared by using one-step RT rPCR mix (AgPath-ID™ One-Step RT-PCR, Thermo, Waltham, MA, USA). Each mix contained 10 µL of Apath-ID RT-PCR buffer 2X, 0.8 µL of Apath-ID RT-PCR enzyme 25X, 10 pm of primers (2 pm for the RNAseP-specific primers), 5 pm of probes, 1 µL of enzyme stabilizer [5,6] (a substrance used to stabilize the enzyme in the prepared RT-rPCR mixes, which was supplied by Nam Khoa Co., Ltd., Ho Chi Minh City, Vietnam), and DNAse/RNAse-free DW at 15 µL. The MPL RT-rPCR mixes were pre-filled into low-profile, white PCR tubes suitable for the real-time PCR instrument used in this study (Biorad’s CFX-96) and kept at −20 °C or lower until used.

### 3.2. Positive Controls

The positive controls of MPL1 and MPL2 were the targeted sequences of the primers and probes prepared previously. The positive controls (except N2_Oligo) were diluted in TE1X to reach the required concentrations of 100 to 1000 copies/5 µL. The N2_Oligo was the positive control for SAR-COV-2 and was used to prepare the standard concentrations to quantify SARS-CoV-2 viral load in the tested samples. The standard concentrations of the N2_Oligo were prepared in TE1X to 3 concentrations of 10^5^, 10^4^, and 10^3^ copies/5 µL. All prepared concentrations of the positive controls and the standards were aliquots in small volumes and stored at −20 °C until used.

### 3.3. Primers for Amplifying and Sanger Sequencing of the Regions Containing Mutations on the S Gene

Based on the GISAID database about mutations on the S gene of the Alpha, Beta, Gamma, Delta, and Omicron variants, nine primer pairs (Table 2) were designed for PCR and then for Sanger sequencing to detect all the mutations that might exist on the S gene.

### 3.4. Performing the MPL RT-rPCR for Detection and Identification of the Variants of SARS-CoV-2

Tested samples: The tested samples were nasopharyngeal swabs taken from people who needed RT-rPCR testing to confirm SARS-CoV-2 infection based on the WHO-guided RT-rPCR protocol for the detection of the E gene [7]. Right after this confirmation test, the samples were kept at −70 °C. In this study, all positive samples and 100 negative samples with SARS-CoV-2 were selected to be tested by the MPL RT-rPCR for the detection and identification of SARS-CoV-2 variants.

Extraction of nucleic acid from test samples: A total of 200 µL of the test samples was taken for nucleic acid extraction by automated method performed on an NK Extractor 96 instrument (Nam Khoa Co., Ltd.), using the DNARNAprep-MAGBEAD kit manufactured by Nam Khoa company. This extraction kit has been studied [8,9,10,11], appraised, and approved by the CDC in Ho Chi Minh City, as well as several provincial CDCs and hospitals that use RT-rPCR testing to confirm SARS-CoV-2.

Real-time PCR running: After extraction, 5 µL of the nucleic acid (NA) extract prepared from the tested samples was added into each of the PCR tube containing 15 µL of the MPL1 or MPL2 RT-rPCR mix. Besides the samples, 5 µL of the positive controls and 5 µL of the negative controls were also added to each of the MPL1 and MPL2 RT-rPCR mix to control the sensitivity and contamination of the real-time PCR when the test was being performed. For each run, 3 MPL2 RT-rPCR mixes were added with 5 µL of the prepared standards for quantification of SARS-CoV-2 viral load in the tested samples. The real-time PCR was run with one 10 min cycle at 45 °C for reverse transcription (RT); then, another 10 min cycle at 95 °C was run to inactivate the RT enzyme and activate the hot start taq polymerase. Finally, forty cycles of real-time PCR including two temperature steps of 95 °C for 15 s and 60 °C for 1 min were run. Four fluorescent channels (FAM, HEX, TexasRED, and CY5) were selected to collect the emitted fluorescent signals. The results were arranged into different patterns, and the variants of SARS-CoV-2 were identified based on the deletion mutations detected in each pattern.

Analysis of the results: (i) The first step was to check the MPL1 and MPL2 that were added with the positive and negative controls. The MLPs added with the negative control must have no amplification signal, indicating that no contamination happened. The MLPs added with the positive control must have the amplification signals with all fluorescent channels at the Ct from 30 to 33, indicating that these MLPs reached the required sensitivity. (ii) After checking sensitivity and contamination, the results of the MPL2 were read. If the TexasRED channel (targeting the N2 of SARS-CoV-2) had no amplification signal and the CY5 channel (targeting the RNAseP) had an amplification signal, the investigator could conclude that the tested sample was negative with SARS-CoV-2. If the TexasRED channel and the CY5 channel had no amplification signal, it could be concluded that the tested sample did not contain enough epithelial cells taken from the nasopharyngeal swab or contained the PCR inhibitors. If the TexasRED channel showed amplification, the investigator could state that the tested sample was positive with SARS-CoV-2, and the viral load was defined by the standard curve given by the MPL2 that was added with 3 standard concentrations of N2_Oligo. (iii) For the sample that was detected [+] with SARS-CoV-2, the investigator continued to read the MPL1 and MPL2 to detect the deletion mutations by target failure. In the MPL1, no amplification in the FAM, HEX, and TexasRED channels indicated, respectively, the presence of the deletion mutation **S:** 142-144 del, **S:** 69-70 del, **S:** N211 del, and **S:** 24-26 del. In the MPL2, no amplification in the FAM and HEX channels indicated the presence of the deletion mutations **S:** 156-157 del and **ORF1a:**141-143 del in the order given.

Identification of SARS-CoV-2 variants: From the deletion mutations detected in the MPL1 and MPL2, SARS-CoV-2 variants could be identified based on the patterns of the target failures that were detected by the MPL RT-rPCR.

### 3.5. Performing Sanger Sequencing for Detection of All Mutations in the S Gene

Sanger sequencing was performed with three steps: The first step was PCR to amplify the nine fragments to be sequenced, followed by sequencing reaction using ABI’s BigDye Terminator and finally sequencing on the ABI 3500 sequencer. To amplify the fragments for Sanger sequencing, we added 5 µL of nucleic acid extracted from the sample to the prepared one-step RT-rPCR mix (which was produced from the “AgPath-ID™ One-Step RT-PCR” as instructed by the manufacturer, and we added 10 pm of forward primer, 10 pm of reverse primer, and DNA/RNA-free DW to the final 15 µL volume); then, one-step RT-PCR was run in the PCR instrument at 45 °C for 10 min for reverse transcription (RT); 95 °C for 10 min for RT enzyme inactivation and activation of the hot start taq polymerase; and, finally, forty cycles of 3 steps including 95 °C for 15 s, 55 °C for 30 s, and 72 °C for 1 min. To prepare the cycle sequencing synthesis, these primers were used with the concentration of 5 pm and were added to the BigDye Terminator v3.1 cycle sequencing kit as instructed by the manufacturer (AB Applied Biosystems, Waltham, MA, USA).

### 3.6. Confirming the Accuracy of the MPL RT-rPCR in the Identification of the Variants

For each pattern of the deletion mutations detected by the MPL RT-rPCR, the samples with high viral load were selected to perform Sanger sequencing. We submitted the S gene sequences detected by Sanger sequencing to the Gisaid website [12] to look up the mutations detected on the S gene of SARS-CoV-2. The deletion mutations detected by Sanger sequencing were compared with those detected by the MPL RT-rPCR in order to decide whether the two results matched. From that, we could confirm the accuracy of the results from the MPL RT-rPCR in the identification of SARS-CoV-2 variants in the tested samples.

### 3.7. Detecting the Limit of Detection (LOD) of the MPL RT-rPCR in the Detection of the Variants

For each of the detected variant, the sample with the highest viral load was chosen, and serial dilutions of the sample (dilution factor of 10) were prepared in TE1X. We then performed nucleic acid extraction and the MPL RT-rPCR to detect the variants in the dilutions of the sample. From the obtained results, the LOD of the variants that could be detected by the MPL RT-rPCR was confirmed.

## 4. Results

### 4.1. The MPL RT-rPCR in the Detection of SARS-CoV-2

From 1 January 2022 to 30 September 2022, 13,970 samples (nasopharyngeal swabs collected from COVID-19 suspected cases) were tested in our laboratory for SARS-CoV-2 detection by using the WHO-guided RT-rPCR protocol for the detection of the E gene. Among these samples, 3239 were positive with SARS-CoV-2 (23.19%). The number of SARS-CoV-2 positive cases that were detected by month were 6.04% (81/1340) on 22 January, 31.01% (569/1835) on 22 February, 39.64% (1920/4843) on 22 March, 23.58% (358/1518) on 22 April, 5.83% (48/823) on 22 May, 2.27% (22/968) on 22 June, 6.06% (71/1172) on 22 July, 11.51% (140/1216) on 22 August, and 11.76% (30/255) on 22 September. After testing the negative and positive samples, the results showed that there were no false positive of SARS-CoV-2 in the 100 SARS-CoV-2 negative samples and no false negative in the 3239 SARS-CoV-2 positive samples. From these results, we could conclude that the MPL RT-rPCR reached 100% sensitivity and specificity for the detection of SARS-CoV-2 directly from the samples.

### 4.2. The MPL RT-rPCR in the Identification of SARS-CoV-2 Variants

After analyzing the results of the MPL RT-rPCR that was performed on 3239 SARS-CoV-2 [+] samples, seven patterns of deletion mutation were found. From these patterns, the detected variants could be identified as follows: *Pattern 1* (Figure 1) was identified as variant **BA.1** because it contained three specific deletion mutations for BA.1: **S:** 142-144 del, **S:** 69-70 del, and **S:** 211 del. *Pattern 2* (Figure 2) was identified as variant **BA.2** because it contained only one specific deletion mutation for BA.2: **S:** 24-26 del. *Pattern 3* (Figure 3) was identified as the recombinant BA.1 and BA.2 (**BA.1/BA.2**) because it contained one specific deletion mutation for BA.1: **S:** 24-26 del, and one specific deletion mutation for BA.2: **S:** 211 del. *Pattern 4* (Figure 4) was identified as variant **BA.4** because it contained three specific deletion mutations for BA.4: **S:** 69-70 del, **S:** 24-26 del, and **ORF1a**:141-143 del. *Pattern 5* (Figure 5) was identified as variant **BA.5** because it contained two specific deletion mutations for BA.5: **S:** 69-70 del and **S:** 24-26 del. *Pattern 6* (Figure 6) and *Pattern 7* (Figure 7) were identified as **Delta** because these patterns contained the specific deletion mutation for delta: **S**: 156-157 del.

The distribution of the variants detected among the 3239 SARS-CoV-2 [+] samples was as follows: BA.1 in 437 cases (13.49%), BA.2 in 2269 cases (70.05%), BA.1/BA.2 in 64 cases (1.98), BA.4 in 7 cases (0.22%), BA.5 in 238 cases (7.35%) and Delta in 224 cases (6.92%).

### 4.3. The Accuracy of MPL RT-rPCR in the Detection of Deletion Mutations

For each of the seven patterns of deletion mutation detected by the MPL RT-rPCR, we selected the sample with the highest viral load to perform Sanger sequencing of the S gene. Table 3 shows the mutations detected in each pattern by Sanger sequencing and a comparison with the deletion mutations detected by the MPL RT-rPCR. As shown in Table 3, in patterns 1, 2, 3, 5, and 6, the deletion mutations detected by the MPL RT-rPCR are similar to those detected by Sanger sequencing. In pattern 4, the deletion mutation **ORF1a**: 141-143 del was not detected by Sanger sequencing because ORF1a did not belong to the S gene and was not sequenced. In pattern 7, as shown in Figure 8 (the histogram of the sequencing), Sanger sequencing detects the **S**: A222V mutation and the **S**:221 mutation with TCG (serine) becoming TCT (serine), and because of these two mutations, the MPL RT-rPCR detects the V211 deletion mutation due to the mismatch of the taqman probe with the target sequence. With these obtained results, we could conclude that the MPL RT-rPCR, in terms of the detection of deletion mutations existing in SARS-CoV-2, reaches the accuracy of Sanger sequencing. Therefore, the identification of SARS-CoV-2 variants by the MPL RT-rPCR is completely comparable with Sanger sequencing of the S gene of the virus.

### 4.4. Limit of Detection (LOD) of MPL RT-rPCR in the Detection of SARS-CoV-2 Variants

By testing the MPL RT-rPCR on the serial dilutions of the sample containing the highest viral load, the LOD of the MPL RT-rPCR in the detection of each of the SARS-CoV-2 variants was defined as the lowest viral load that the deletion mutation pattern of the variants was detected by the MPL RT-rPCR. The results demonstrated that the LOD (copies/reaction volume) of the MPL RT-rPCR in the detection of the variants ranged from 1 to 10 copies for Omicron BA.2 and BA.5, and from 10 to 100 copies for Delta, Omicron BA.1, BA.1/BA.2, and BA.4. It means that the LOD of the MPL RT-rPCR in the detection of BA.2 and BA.5 is 100 to 1000 copies/mL of the samples and from 1000 to 10,000 copies/mL of the samples for Delta, BA.1, BA.4, and BA.1/BA.2. Besides the determination of the LOD, we also attempted to find the lowest viral load of each of the variants existing in the tested samples that were detected by the MPL RT-rPCR. The obtained results also showed no difference with the LOD results. It was noted that the viral loads were calculated via the quantification of the viral N gene of SARS-CoV-2 existing in the tested samples based on the standard curve given by three standard concentrations (S1, S2, and S3).

### 4.5. Distribution of the Detected SARS-CoV-2 Variants

The SARS-CoV-2 variants that were detected and identified from the 3239 SARS-CoV-2 [+] samples by the MPL RT-rPCR are shown in Table 4. By analyzing these results, we could summarize some main points: (i) The Delta variant took up 96.3% on 22 January, then dropped down to 14.4% on 22 February, and nearly disappeared from 22 June onward. (ii) Omicron BA.1 started to appear on 22 January with a ratio of 3.7%, reached a maximum ratio of 32.7% on 22 February, and then gradually decreased to 4.2% on 22 May and disappeared from 22 June onward. (iii) Omicron BA.2 emerged from 22 February onward with a high ratio of 52.7%, peaked at 87.4% on 22 April, declined to 23.9% on 22 July, and remained at a moderate ratio of 10% on 22 September. (iv) The recombinant variants of BA.1 and BA.2 (BA.1/BA.2) appeared with a low ratio (0.2%) on 22 February, peaked at 2.9% on March, dropped to 2% and 2.1% on 22 April and 22 May, and was then totally unreported from 22 June onward. (v) Omicron BA.4 stared to appear on 22 July, with a ratio of 4.2% on 22 July and 3.6% on 22 August, and disappeared on 22 September. (vi) Omicron BA.5 only took up 0.3% of the cases on 22 March, increased to 68.3% and 70.4% on June and 22 July, and reached its maximum ratio of 94.3% on 22 August and 90% on 22 September.

## 5. Discussion

According to the World Health Organization (WHO) and ECDC, in order to detect variants of SARS-CoV-2 and depending on the purpose, laboratories may choose the appropriate technical solution [13]. Whole-genome viral sequencing (WGS) solution based on next-generation sequencing (NGS) technology [14,15,16], or partial or complete sequencing of the S gene based on NGS or Sanger technology [14,15,16], is the solution to reaffirm the detected variation in screening tests or to monitor the appearance of new variants or sub-variants [17,18]. However, sequencing can only be performed at centers with corresponding equipment, and the time required to get the results is often long because it is necessary to wait for sufficient number of tested samples for one sequencing set (especially NGS). For that reason, it is not possible to apply NGS in diagnostic or in small epidemic control laboratories.

Currently, with the rapid spread of Omicron variants, the need to detect variants is dire in diagnostics and in epidemic control. Diagnosis is needed since Omicron variants may be resistant to monoclonal antibodies for therapeutic use; hence, if the diagnostic result identifies a patient with COVID-19 due to Omicron variants, physicians will not prescribe monoclonal antibodies, and if necessary, will only use antiviral drugs for specific treatment. It is necessary in epidemic control because Omicron variants spread faster, are resistant to neutralizing antibodies produced by the body through vaccination or natural infection, and do not have the potency for lower respiratory tract damage [19]. Consequently, constant monitoring of current circulating variants, including Omicron, is necessary to timely change epidemic control solutions. In response to these requests, the WHO and CDC have listed several commercialized test kits for the detection of Omicron variants, such as Roche’s TIB MolBiol, which detects the three mutations of S371L/S373P, 214insEPE, and E484A [20,21]; Thermo Fisher TaqPath kit, which detects H69del and V70del mutations [18]; and Seegene kit, which detects E484A, N501Y, H69del, and V70del mutations [22] simultaneously. In addition, the WHO has also introduced procedures developed by some research institutes to detect Omicron variants and other variants of SARS-CoV-2 [23,24,25,26,27]. In general, commercialized test kits or procedures developed to detect Omicron variants as well as other variants are based on RT real-time PCR technology, either based on the analysis of the melting curve of the amplification product, the detection of S gene target failure (SGTF), the detection of specific single nucleotide polymorphisms (SNPs) of the variants, or molecular clamping based on the insertion of a xeno nucleic acid (XNA) to lock the non-mutated sequence [28].

In this work, we designed and applied the MPL RT-rPCR procedure based on target-failure real-time PCR for the detection and identification of current circulating variants of SARS-CoV-2, and from the results that we received and analyzed in this study, we can conclude four advantages as shown in the obtained results: (i) With the usage of the primers and taqman probes to detect four deletion mutations on the S gene and one deletion mutation on the ORF1a gene, this MPL RT-rPCR could detect and identify current circulating SARS-CoV-2 variants in Vietnam and possibly in the world (including Delta, Omicron BA.1, Omicron BA.2, Omicron BA.4, and Omicron BA.5) directly from samples taken from COVID-19 suspected individuals. (ii) With the usage of the primers and taqman probe specific for the human RNAse-P gene, this MPL RT-rPCR could control the quality of the tested samples and the quality of the nucleic acid extraction step to avoid false negative result due to poor sample quality or the low efficacy of the nucleic acid extraction step. (iii) With the use of the primers and probe specific for the conserved region N2 of the N gene of SARS-CoV-2, this MPL RT-rPCR could quantify SARS-CoV-2 viral load based on the standard curve resulting from the real-time PCR of the three standards included in the test. (iv) With the possibility of detecting and identifying current circulating variants at a low LOD (1–100 copies/volume of reaction), this MPL RT-rPCR could be useful in diagnosis as well as in monitoring the distribution of SARS-CoV-2 variants since it could be applied even on samples with low SARS-CoV-2 viral load, which sequencing could not achieve. However, we also acknowledge that this MPL RT-rPCR could not identify BA.2.12.1 (22C), BA.2.75 (22D), BQ.1 (22E), and XBB (22F) since these variants have completely similar deletion mutations to BA.2 (BA.2.12.1, BA.2.75, and XBB) or BA.5 (BQ.1), and relatively few other mutations distinct from BA.2 or BA.5. However, these variants have rarely been reported in Vietnam as well as other regions worldwide, and with our existing situation, these variants will probably not be a serious threat because they seem to be unable to outgrow the current BA.5.

The WHO and ECDC have also recommended that if it is desirable to sequence a portion of the S gene using Sanger sequencing to detect the variants of SARS-CoV-2, laboratories should sequence a fragment of the gene that covers the entire N-terminus (N terminal) and RBD region of the virus (amino acid from 1 to 541 equivalent to 1623 bps) [13], and, ideally, covers the additional S1/S2 region, i.e., up to 800 amino acids (equivalent to 2400 bps). In this study, to ensure the sensitivity of Sanger sequencing, we designed and applied nine primer pairs to amplify nine fragments of the S gene that were reported by https://covariants.org (Enabled by data from GISAID; accessed on 3 March 2023) [3] to discover all the mutations associated with the occurrence of SARS-CoV-2 variants. These segments range in size from 200 to 350 bps, so the sequencing region covers both the N-terminus of the S, RBD, and S1/S2 genes. The aim of this study was to use the results of the sequencing of these nine fragments as a standard to reaffirm the results of the MPL RT-rPCR in the identification of variants without the need to send the samples for sequencing using NGS technology. Moreover, with Sanger sequencing technology and only sequencing these nine fragments of this S gene, a diagnostic laboratory or epidemiological research can detect the emergence of new variants without the need to use next-generation sequencing technology that requires not only high cost but also bioinformatics knowledge to analyze the results [13].

Omicron variants carry 15 mutations in the receptor-binding domain (RBD), and five of them (G339D, N440K, S477N, T478K, and N501Y) were reported to help the virus increase its adhesion to ACE2 receptors on human respiratory epithelial cells [29]. In addition, near the S1/S2 cleavage site, Omicron has three mutations (H655Y, N679K, and P681H) that make it easier for the virus to release its RNA into the cell cytoplasm after endocytosis [30,31]. These eight mutations help Omicron variants become more contagious than Delta variants and other VOC variants [29,30,31]. All these mutations were found in the Omicron variants (BA.1, BA.2, BA.1/BA.2, BA.4, and BA.5) that were sequenced in this study. On the RBD, Omicron variants also have seven mutations (K417N, G446S, E484A, Q493R, G496S, Q498R, and N501Y) that are found to be resistant to antibodies that the body produces after natural infection or vaccination to neutralize the virus’ ability to bind to ACE2 receptor [32,33,34,35,36]. The sequencing results of our Omicron variants in this study show that Omicron BA.1 misses the K417N mutation; Omicron BA.2 and BA.1/BA.2 miss the G446S and G496S mutation; and BA.4 and BA.5 miss the G446S, Q493R, and G496S mutations. However, the missing of these mutations on these Omicron variants may not affect the ability of these variants to infect vaccinated or even naturally infected people since, in Vietnam, a lot of people, who were already infected in the epidemic caused by the Delta variant and/or vaccinated with two, three or even four shots, still get reinfected with the Omicron variants.

On the N-terminal region (NTD) [37] of the S gene of Omicron variants, four missense mutations (A67V, T95I, G142D, and L212I) [32,33,34], six deletion mutations (H69del, V70del, V143del, Y144del, Y145del, and N211del) [33,34,35], and an insertion mutation (ins214EPE) [32,33,34] have been reported and predicted to increase the resistance of Omicron to monoclonal antibodies used in current treatment, such as bamlanivimab plus etesevimab or casirivimab plus imdevimab [36,38,39,40]. Compared to the sequencing results of the Omicron variants in this study, on Omicron BA.1, we did not find mutation A67V; On Omicron BA.2, the mutations A67V, T95I, L212I, H69del, V70del, V143del, Y144del, Y145del, N211del, and 214insEPE were not found; on Omicron BA.1/BA.2, the mutations A67V, T95I, H69del, V70del, V143del, Y144del, and Y145del, were not found; and on Omicron BA.4 and BA.5, the mutation A67V, T95I, L212I, V143del, Y144del, Y145del, N211del, and 214insEPE were not found. Considering the mutations which are responsible for Omicron’s resistance to monoclonal antibodies used in treatment, Omicron BA.1 in Vietnam is likely to be more resistant than other variants, such as BA.2, BA.1/BA.2, BA.4, and BA.4. However, we are unable to provide clinical evidence for that, since in Vietnam, monoclonal therapy is not a common treatment of COVID-19, especially during this time when severe cases caused by Omicron variants are scarce.

According to the Vietnam coronavirus stats from the COVID-19 Data Repository collected by the Center for Systems Science and Engineering at Johns Hopkins University (JHU CSSE COVID-19 Data), from 22 January to 22 September, there were two periods that the number of new COVID-19 cases in Vietnam increased: the 3-month period from 22 February to end of 22 April and the first two-week period of 22 August. In our study, we also received the same results: The number of COVID-19 positive cases that we detected by RT-rPCR were high from 22 February to April, with the highest on 22 March, decreased from 22 June to 22 July, and slightly increased on 22 August. This might be connected with the analysis of the distribution of SARS-CoV-2 variants by month that we received in this study: In 22 January, the Delta variant was still the main variant, although Omicron BA.1 also started to appear. From 22 February to 22 May, Omicron BA.2 dominated. However, from 22 June to 22 September, Omicron BA.5 had replaced Omicron BA.2 to become the main circulating variant.

The Omicron variants BA.1 and BA.2 with more than 30 mutations in the S gene began to appear in South Africa in November 2021 with a fairly fast transmission rate, surpassing the previous Delta variant to gradually replace this variant worldwide [1,2]. From January to February 2022, in South Africa, BA.4 and BA.5, which are variants derived from BA.2, began to appear [41] and spread even faster than BA.2, probably because of two special mutations, L452R and F486V, that help them attach and enter cells faster [42,43]. In our study, the results show that Omicron BA.1 started appearing from mid-January in 2022. Then, from mid-February, Omicron BA.2 gradually replaced the BA.1 variant. Omicron BA.5 appeared sporadically from March to May, and from June to September, it began to be the predominant variant, almost replacing BA.2. Omicron BA.4 variant started to appear with a few cases in July and August. With these results, we found that no more than three months after the first appearance in South Africa, BA.1, BA.2, BA.5, and BA.4 already existed in Vietnam. In addition, we also note that BA.4 cases were rare. Although BA.4 and BA.5 have similar mutations in the S gene, the reason why BA.4 does not spread as quickly as BA.5 is still unclear [44]. One of the reasons that we suggest is that it is probably related to the ORF1a:141-143del mutation that presents on BA.4 but not on BA.5 [3].

## 6. Conclusions

Although having appeared only since the end of November 2021, Omicron variant is now considered to be the predominant variant worldwide and in Vietnam due to its presumably 2.8 times faster spreading rate than the previous Delta variant [45]. Not only that, the Omicron variant also evolved into four sub-variants, BA.1, BA.2, BA.4 and BA.5, with BA.2 spreading 1.5 times faster than BA.1 [45,46] and 4.2 times faster than Delta variant [45,46], and the recent Omicron BA.5 was reported to transmit even faster than BA.2 and, thus, gradually become predominant in the world. Fortunately, because the mechanism of cell entry is only through ACE2 receptor, and not in combination with TMPRSS2 receptor, Omicron is thought to be less lung invasive than the Delta variant [47]. Besides, with the current wide vaccine coverage in Vietnam and various countries around the world, Omicron is more likely to cause mild disease than Delta or previous variants. However, continuous monitoring of the incidence of different sub-variants of Omicron is essential not only in epidemic control but also in diagnostic monitoring. The real-time PCR approach is considered most suitable for this purpose as there are numerous laboratories at disease control centers, as well as at hospitals, that are now equipped with real-time PCR facilities. Therefore, this study is dedicated to this purpose. Hopefully, this real-time PCR method will be applied by a great number of laboratories due to its feasibility, easy implementation, simple analysis of results, and high sensitivity.

## Figures and Tables

**Figure 1 diagnostics-13-01364-f001:**
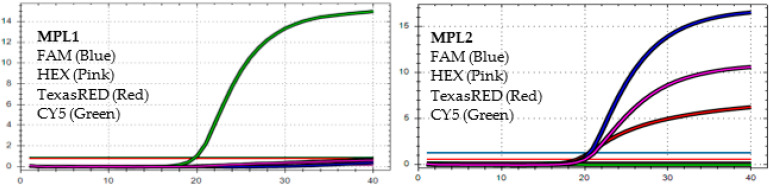
*In MPL1*, the target failures are detected in 3 channels, FAM, HEX, and TexasRED. *In MPL2*, no target failure is detected in 3 channels, FAM, HEX, and TexasRED. No amplification is detected in channel CY5 because of the competition of high viral load with the IC (human RNAse P). This pattern is identified as variant **BA.1**.

**Figure 2 diagnostics-13-01364-f002:**
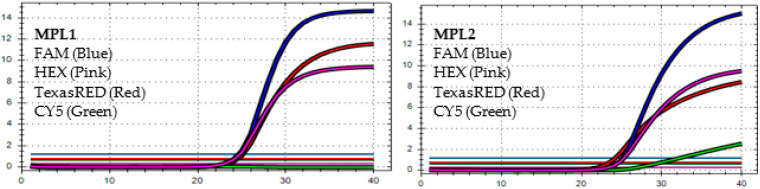
*In MPL1*, the target failure is detected in channel CY5. *In MPL2*, no target failure is detected in 3 channels, FAM, HEX, and TexasRED. The amplification detected in channel CY5 is weak because of the competition of high viral load with the IC (human RNAse P). This pattern is identified as variant **BA.2**.

**Figure 3 diagnostics-13-01364-f003:**
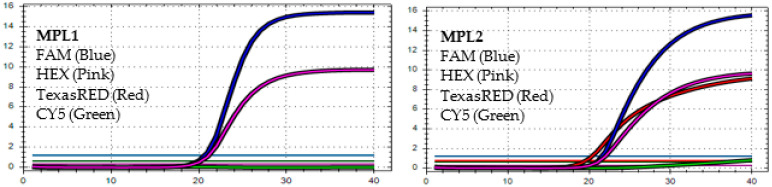
*In MPL1*, the target failures are detected in 2 channels, TexasRED and CY5. *In MPL2*, no target failure is detected in 3 channels, FAM, HEX, and TexasRED. No amplification is detected in channel CY5 because of the competition of high viral load with the IC (human RNAse P). This pattern is identified as the recombination of **BA.1** and **BA.2 (BA.1/BA.2)**.

**Figure 4 diagnostics-13-01364-f004:**
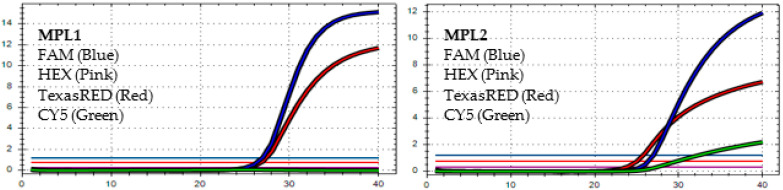
*In MPL1*, the target failures are detected in 2 channels, HEX and CY5. *In MPL2*, the target failure is detected in channel HEX. This pattern is identified as variant **BA.4**.

**Figure 5 diagnostics-13-01364-f005:**
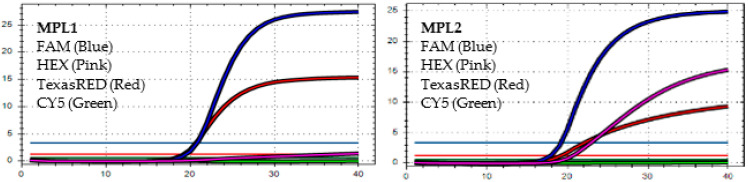
*In MPL1*, the target failures are detected in 2 channels, HEX and CY5. *In MPL2*, no target failure is detected in 3 channels, FAM, HEX, and TexasRED. No amplification is detected in channel CY5 because of the competition of high viral load with the IC (human RNAse P). This pattern is identified as variant **BA.5**.

**Figure 6 diagnostics-13-01364-f006:**
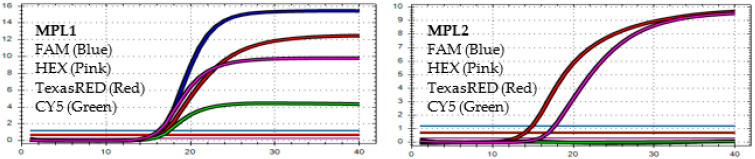
*In MPL1*: No target failure is detected. *In MPL2*, the target failure is detected in channel FAM. No amplification is detected in channel CY5 because of the competition of high viral load with the IC (human RNAse P). This pattern is identified as variant **Delta**.

**Figure 7 diagnostics-13-01364-f007:**
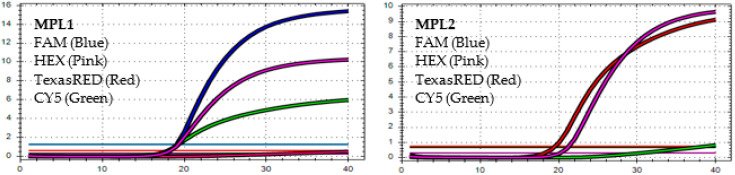
*In MPL1*: The target failure is detected in TexasRED channel. *In MPL2*, the target failure is detected in channel FAM. No amplification is detected in channel CY5 because of the competition of high viral load with the IC (human RNAse P). This pattern is identified as variant **Delta**.

**Figure 8 diagnostics-13-01364-f008:**
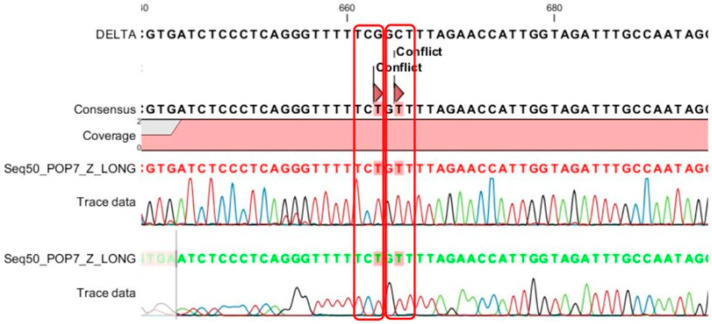
Sanger sequencing of the 7 patterns in the samples detects 2 mutations, A222V and (TCG)221(TCT), and because of these two mutations, mismatch happens between the 222A_tqPR (taqman probe) with the target sequence.

**Table 1 diagnostics-13-01364-t001:** The sequences of the designed primers, probes, and positive controls specific for the different deletion mutations.

	**Name**	**Sequence**	**Targets**
**MPL1**	142-144_TqF	TGTAATGATCCATTTTTGGGTGTT	**S:** 142-144 del **(Existed in BA.1)**
	142-144_TqR	AAACTCTGAACTCACTTTCCATCCA	
	142-144_TqPR	FAM-TACCACAAAAACAACAAAAG-MGBNFQ	
	69-70_TqF	TGTTCTTACCTTTCTTTTCCAATGTT	**S:** 69-70 del **(Existed in BA.1, BA.4, BA.5)**
	69-70_TqR	AGGACAGGGTTATCAAACCTCTTAGT	
	69-70_PrDEL[−]	HEX-CATGCTATACATGTCTCTGGGACCAATGG-SFCQ1	
	222AV_TqF	ACCTAGTGATGTTAATACCTATTGGCA	**S:** 211 del and Ins214EPE **(Existed in BA.1)**
	222AV_TqR	CGCCTATTAATTTAGTGCGTGATC	
	222A_tqPR	TexasRED-ACCAATGGTTCTAAAGCCGAAAAACCCT-BHQ2	
	24-26_TqF	GCCACTAGTCTCTAGTCAGTGTGTTAAT	**S:** 24-26 del **(Existed in BA.2, BA.4, BA.5)**
	24-26_TqR	GTCAGGGTAATAAACACCACGTGT	
	24-26_TqPR	Cy5-CCAGAACTCAATTACCCCCTGCATACACT-BHQ3	
**MPL2**	156to158wt_TqF	AAGTTGGATGGAAAGTGAGTTCAGA	**S:** 156-158 del **(Existed in Delta)**
	156to158_TqR	TCCATAAGAAAAGGCTGAGAGACA	
	156to158_TqPR	FAM-TCTAGTGCGAATAATTG-MGBNFQ	
	ORF1a_TqF	CGGTAATAAAGGAGCTGGTGG	**ORF1a:** 141-F143 del **(Existed in BA.4)**
	ORF1a_TqR	ATCTTCATAAGGATCAGTGCCAAG	
	ORF1a_TqPR	HEX-TCGCCTAAGTCAAATGACTTTAGATCGGC-SCFQ1	
	N2-F2019-nCoV	TTACAAACATTGGCCGCAAA	N2 of SARS-CoV-2
	N2-R2019-nCoV	GCGCGACATTCCGAAGAA	
	N2Probe	5′-TexasRED-ACAATTTGCCCCCAGCGCTTCAG-BHQ2-3′	
	RNAseP_TQ_F	AGATTTGGACCTGCGAGCG	Human RNAseP
	RNAseP_TQ_R	GAGCGGCTGTCTCCACAAGT	
	HuRnaseP-Pr	CY5-TTCTGACCTGAAGGCTCTGCGCG-BHQ3	

**Table 2 diagnostics-13-01364-t002:** The sequences of the primers for amplification and then Sanger sequencing of the 9 fragments of the S gene that might contain the reported mutations.

Name: Sequence		Seq No (Size)
6F: TGTTTTTCTTGTTTTATTGCCACTA	330R: CAAAAATCCAGCCTCTTATTATGT	Seq1 (325 bps)
288F: GAAGTCTAACATAATAAGAGGCTGG	628R: TAGGCGTGTGCTTAGAATATATTTT	Seq2 (341 bps)
602F: TTAAAATATATTCTAAGCACACGCC	814R: GTTGAAGATAACCCACATAATAAGC	Seq3 (213 bps)
987F: TCCTAATATTACAAACTTGTGCCCT	1295R: CAGCCTGTAAAATCATCTGGTAAT	Seq4 (309 bps)
1279F: GATGATTTTACAGGCTGCGTTAT	1563R: TGGTGCATGTAGAAGTTCAAAAGA	Seq5 (285 bps)
1541F: CTTTTGAACTTCTACATGCACCAG	1877R:GCATGAATAGCAACAGGGACTT	Seq6 (337 bps)
1933F: ACACGTGCAGGCTGTTTAATAG	2189R: GACACTGGTAGAATTTCTGTGGTAA	Seq7 (257 bps)
2261F: TGCAATATGGCAGTTTTTGTACA	2604R: TTCATCTGTGAGCAAAGGTGG	Seq8 (344 bps)
2788F: GCTATTGGCAAAATTCAAGACTC	3111R: TGATTGTCCAAGTACACACTCTGA	Seq9 (324 bps)

**Table 3 diagnostics-13-01364-t003:** The mutations detected in each pattern by Sanger sequencing and compared with the deletion mutations detected by the MPL RT-rPCR.

	Mutation Detected by Sanger Sequencing of S Gene of SARS-CoV-2	Deletion Mutation Detected by the MPL RT-rPCR
**Pattern 1.**	**H69del, V70del**, T95I, G142D, **V143del**, **Y144del**, **Y145del**, **N211del**, L212I, ins214APE, P217T, G339D, R346K, S371L, S373P, S375F, N440K, G446S, S477N, T478K, E484A, Q493R, G496S, Q498R, N501Y, Y505H, T547K, D614G, H655Y, N679K, P681H, N764K, D796Y, N856K, Q954H, N969K, L981F	**S**: **69-70 del**, **S**: **142-144 del**, **S**: **N211del**
**Pattern 2**	T19I, **L24del**, **P25del**, **P26del**, A27S, G142D, G339D, S371F, S373P, S375F, T376A, D405N, K417N, N440K, S477N, T478K, E484A, Q493R, Q498R, N501Y, Y505H, D614G, H655Y, N679K, P681H, N764K, D796Y, N856K, Q954H, N969K	**S**: **24-26 del**
**Pattern 3**	T19I, **L24del**, **P25del**, **P26del**, A27S, G142D, **N211del**, L212I, V213G, G339D, S371F, S373P, S375F, T376A, D405N, K417N, N440K, S477N, T478K, E484A, Q493R, Q498R, N501Y, Y505H, D614G, H655Y, N679K, P681H, N764K, D796Y, Q954H, N969K.	**S**: **24-26 del**, **S**: **211 del**
**Pattern 4**	T19I, **L24del**, **P25del**, **P26del**, A27S, **H69del**, **V70del**, G142D, G339D, S371F, S373P, S375F, T376A, D405N, R408S, K417N, N440K, L452R, S477N, T478K, E484A, F486V, Q498R, N501Y, Y505H, D614G, H655Y, N679K, P681H, N764K, D796Y, Q954H, N969K	**S**: **24-26 del**, **S**: **69-70 del**, **ORF1a**: **141-143 del***
**Pattern 5**	T19I, **L24del**, **P25del**, **P26del**, A27S, **H69del**, **V70del**, G142D, G339D, S371F, S373P, S375F, T376A, D405N, R408S, K417N, N440K, L452R, S477N, T478K, E484A, F486V, Q498R, N501Y, Y505H, D614G, H655Y, N679K, P681H, N764K, D796Y, Q954H, N969K	**S**: **24-26 del**, **S**: **69-70 del**
**Pattern 6**	T19I, G142D, E156G, **F157del**, **R158del**, A222V, L452R, T478K, D614G, P681H, D950N	**S**: **156-157 del**
**Pattern 7**	T19R, G142D, E156G, **F157del**, **R158del**, A222V, L452R, T478K, D614G, P681H, D950N	**S**: **156-157 del**, **S**: **N211del**

The deletion mutation **ORF1a**: 141-143 del was not detected by Sanger sequencing because ORF1a was not sequenced.

**Table 4 diagnostics-13-01364-t004:** The distribution of the SARS-CoV-2 variants detected by the MPL RT-rPCR among 3239 SARS-CoV-2 [+] samples collected from January to September 2022.

	BA.1	BA.2	BA.1/BA.2	BA.4	BA.5	Delta	[+] Cases
	N (%)	N (%)	N (%)	N (%)	N (%)	N (%)	N
January 2022	3 (3.70)	0 (0.00)	0 (0.00)	0 (0.00)	0 (0.00)	78 (96.30)	81
February 2022	186 (32.69)	300 (52.72)	1 (0.18)	0 (0.00)	0 (0.00)	82 (14.41)	569
March 2022	229 (11.93)	1592 (82.92)	55 (2.86)	0 (0.00)	5 (0.26)	39 (2.03)	1920
April 2022	17 (4.75)	313 (87.43)	7 (1.96)	0 (0.00)	7 (1.96)	14 (3.91)	358
May 2022	2 (4.17)	34 (70.83)	1 (2.08)	0 (0.00)	1 (2.08)	10 (20.83)	48
June 2022	0 (0.00)	7 (31.82)	0 (0.00)	0 (0.00)	15 (68.18)	0 (0.00)	22
July 2022	0 (0.00)	17 (23.94)	0 (0.00)	3 (4.23)	50 (70.42)	1 (1.41)	71
August 2022	0 (0.00)	3 (2.14)	0 (0.00)	4 (2.86)	133 (95.00)	0 (0.00)	140
September 2022	0 (0.00)	3 (10.00)	0 (0.00)	0 (0.00)	27 (90.00)	0 (0.00)	30
Total	437 (13.49)	2269 (70.05)	64 (1.98)	7 (0.22)	238 (7.35)	224 (6.92)	3239

## Data Availability

Data can be found at University of Bari-Italy and Phan Chau Trinh University of MedicineVietnam.

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
