# Peer review of "Multiplex RT Real-Time PCR Based on Target Failure to Detect and Identify Different Variants of SARS-CoV-2: A Feasible Method That Can Be Applied in Clinical Laboratories"

_diagnostics, 2023, doi:10.3390/diagnostics13081364_

Round 1

Reviewer 1 Report

Report:

Multiplex RT real-time PCR based on the target failure to de-

tect and identify the different variants of SARS-COV-2: a feasi-

ble method that can be applied in the clinical laboratories

Minor Comments:

The study is really interesting and can be further improved. Please adjust correctly the captions and the images. Both need attention.

Major comments:

1:

Kindly specify in your abstract either:

The variants studied are the variants of interest,

Or

Variants of consequence

Or

Both.

2:

For sanger sequencing, lots of libraries are developed using R and Python.

Please mention in acknowledgment, if any of such ready to use algorithms were adopted.

Please give a schematic of the implementation of the genetic sequencing data analysis prior to table 1.

After these modifications, I think that the current work can contribute positively to the research of COVID-19.

Author Response

Major comments:

1:

Kindly specify in your abstract either:

The variants studied are the variants of interest,

Or

Variants of consequence

Or

Both.

Thanks for the comments we pursued Variants of Concern 

as precisely stated in the abstract we've seen a number of the same genetic mutations emerging similarly to different scientists around the world. Furthermore, we can assert an increase in transmissibility, more severe disease (e.g. increased hospitalizations or deaths), a significant reduction in neutralization by antibodies generated during previous infection or vaccination, reduced effectiveness of treatments or vaccines, or diagnostic detection failures, "according to the CDC

2:

For sanger sequencing, lots of libraries are developed using R and Python.

Please mention in acknowledgment, if any of such ready-to-use algorithms were adopted.

We have proceeded as per the reviewer's suggestions

Please give a schematic of the implementation of the genetic sequencing data analysis prior to table 1.

We have proceeded as per the reviewer's suggestions

Reviewer 2 Report

Multiplex RT real-time PCR based on the target failure to detect and identify the different variants of SARS-COV-2: a feasible method that can be applied in the clinical laboratories

 The authors developed and evaluated a Multiplex RT real-time PCR (MPL RT-rPCR) based on the principle of target failure to detect and identify the different variants of SARS-COV-2. They state that this real-time PCR procedure will be used by many clinical laboratories due to its ease as well as the feasibility of implementation and its high sensitivity.

 It is an interesting approach. The work is of interest because it may allow a faster identification of SARS-CoV-2 variables.

However, some points need attention:

- The text is sometimes in the past, sometimes it is in the present, it must standardize.

- Some points of the materials and methods are too descriptive, especially in already standardized techniques, and such a description is unnecessary

- In subchapter 4.1 sentence “One hundred samples [-] with SARS-COV-2 and all of 3239 samples [+] with SARS-COV-2 and were selected to be tested with MPL RT-rPCR. ” does not make sense, needs to be rewritten.

- Figures need to be identified, numbered, and need to be referred to in the text before they appear. It doesn't always happen.

- Graphs with different patterns are figures, they should be labelled as figure.

- Tables should only appear after being mentioned in the text, which is not always the case.

- Photo1 must be called figure, and in this case, it will be figure 2. The caption must follow the figure and not after the text.

- The differences (namely the advantages) between this approach and the various existing kits and techniques for detecting variants must be emphasized and clearly stated.

Author Response

The text is sometimes in the past, sometimes it is in the present, it must be standardize.

  • We proceeded as per the reviewer's comment. It can be seen in the text all the changes we made.
  • Some points of the materials and methods are too descriptive, especially in already standardized techniques, and such a description is unnecessary.
  • We proceeded as per the reviewer's comment. It can be seen in the text all the changes we made.
  • In subchapter 4.1 sentence “One hundred samples [-] with SARS-COV-2 and all of 3239 samples [+] with SARS-COV-2 and were selected to be tested with MPL RT-rPCR. ” does not make sense, needs to be rewritten.
  • We proceeded as per the reviewer's comment. It can be seen in the text all the changes we made.
  •  
  • Figures need to be identified, numbered, and need to be referred to in the text before they appear. It doesn't always happen.
  • We proceeded as per the reviewer's comment. It can be seen in the text all the changes we made.
  •  
  • Graphs with different patterns are figures, they should be labelled as figure.
  • We proceeded as per the reviewer's comment. It can be seen in the text all the changes we made.
  • Tables should only appear after being mentioned in the text, which is not always the case.
  • We proceeded as per the reviewer's comment. It can be seen in the text all the changes we made.
  • Photo1 must be called figure, and in this case, it will be figure 2. The caption must follow the figure and not after the text.
  • We proceeded as per the reviewer's comment. It can be seen in the text all the changes we made.
  • The differences (namely the advantages) between this approach and the various existing kits and techniques for detecting variants must be emphasized and clearly stated.
  • We proceeded as per the reviewer's comment. It can be seen in the text all the changes we made.

Round 2

Reviewer 2 Report

Multiplex RT real-time PCR based on the target failure to detect and identify the different variants of SARS-COV-2: a feasible method that can be applied in the clinical laboratories

The authors responded acceptably to the questions posed. There are two sentences that still need to be corrected:

3.4. “In this study, all of the positive samples and 100 samples negative samples with SARS-COV-2 …” should be “ In this study, all of the positive samples and 100 negative samples for SARS-COV-2 … “

4.3. “In pattern 7, as shown in picture 8…” should be figure 8